# A Novel Deep Learning Model for the Detection and Identification of Rolling Element-Bearing Faults

**DOI:** 10.3390/s20185112

**Published:** 2020-09-08

**Authors:** Alex Shenfield, Martin Howarth

**Affiliations:** National Centre for Excellence in Food Engineering, Sheffield Hallam University, Sheffield S1 1WB, UK; m.howarth@shu.ac.uk

**Keywords:** condition monitoring, fault diagnosis, deep learning, artificial intelligence

## Abstract

Real-time acquisition of large amounts of machine operating data is now increasingly common due to recent advances in Industry 4.0 technologies. A key benefit to factory operators of this large scale data acquisition is in the ability to perform real-time condition monitoring and early-stage fault detection and diagnosis on industrial machinery—with the potential to reduce machine down-time and thus operating costs. The main contribution of this work is the development of an intelligent fault diagnosis method capable of operating on these real-time data streams to provide early detection of developing problems under variable operating conditions. We propose a novel dual-path recurrent neural network with a wide first kernel and deep convolutional neural network pathway (RNN-WDCNN) capable of operating on raw temporal signals such as vibration data to diagnose rolling element bearing faults in data acquired from electromechanical drive systems. RNN-WDCNN combines elements of recurrent neural networks (RNNs) and convolutional neural networks (CNNs) to capture distant dependencies in time series data and suppress high-frequency noise in the input signals. Experimental results on the benchmark Case Western Reserve University (CWRU) bearing fault dataset show RNN-WDCNN outperforms current state-of-the-art methods in both domain adaptation and noise rejection tasks.

## 1. Introduction

Recent advances in the Industrial Internet-of-Things have meant that real-time acquisition of large amounts of machine operating data is now increasingly common. This data can inform many aspects of industrial plant management—from high-level process optimization across a factory (or even across multiple sites) to monitoring the performance of individual machines. However, of critical importance within the manufacturing industry is the ability for this data to provide remote condition monitoring of machinery. In many situations, the breakdown of a machine has a significant impact not only on the production capacity and operating costs of a factory but also on employee safety.

Rolling element bearings are one of the key components in many types of industrial machinery, including mechanical power transmission systems and other forms of electromechanical drive systems, and degradation of these has a significant impact on both the performance and lifespan of a system. Several studies [1,2,3] have estimated that between 40% and 70% of electromagnetic drive system failures are due to bearing faults. Vibration data is commonly used in the detection and identification of faults in rolling element bearings, with bearing damage producing impulse responses in the vibration signal where bearing components contact the fault area [4]. Envelope analysis [5] is often used as the primary tool for diagnosis of bearing faults from these vibration signals; however, it often requires extensive domain knowledge (such as rotating shaft speed and load angle for the specific machine being analyzed) and its accuracy can be significantly affected by factors such as signal noise, changes in operating conditions, and rotational slips in the system.

In the past 10–15 years, there has been substantial interest in intelligent data-driven fault diagnosis techniques that overcome many of the limitations of envelope analysis based methods. Conventional machine learning approaches to this data-driven fault diagnosis process consists of three main stages:Feature extraction—where meaningful features are extracted from the raw time series data using signal processing techniques.Feature selection—where the dimensionality of the features extracted in the previous step is reduced to the most important features.Classification—where the selected features are used to identify the class of the fault in the system.

For vibration-based fault detection and diagnosis, there are a range of features from both the frequency domain and the time domain that are commonly used to characterize the vibration signals; including signal amplitudes, kurtosis, and power spectral densities. These features have been used in the literature to identify bearing faults using machine learning methods such as Artificial Neural Networks (ANNs) [6], Support Vector Machines (SVMs) [7,8,9], and k-Nearest Neighbor classifiers [10]. However, a key drawback with this approach is that these handcrafted features have to be decided a priori (often by trial and error) and the success of the classification algorithm is highly dependent on the choice of features used as an input. Many of the existing intelligent data-driven fault diagnosis techniques in the literature struggle to adapt to different working conditions (such as different motor loads, different rotating shaft speeds, or varying amounts of environmental noise). This is due in part to the features extracted in one domain not necessarily transferring optimally to a different domain.

In recent years, deep learning approaches have been shown to consistently outperform other techniques on a range of natural language processing and computer vision tasks [11,12]. One of the most significant advantages of applying deep learning approaches is their capability for end-to-end learning without the need for complex feature engineering steps. The ability of deep learning algorithms to automatically extract information of interest from the raw data not only helps ensure robust decision making, but also makes their use more feasible in many domains.

In this paper, we propose a novel deep learning model for fault detection and diagnosis of time-series data that addresses many of the problems with existing intelligent data-driven fault diagnosis techniques. The key contributions of this paper are:We propose a novel deep learning framework for fault detection on time-series data that combines elements of convolutional neural networks (CNNs) with a recurrent neural network (RNN) pathway.We show that our proposed framework works directly on the raw temporal data obviating the need for manual feature extraction or noise removal.We show that our framework exhibits excellent robustness to both environmental noise and changes in operating conditions.We compare our framework to several state-of-the-art approaches using both conventional machine learning techniques and deep learning approaches.

The rest of this paper is organized as follows: Section 2 will provide a brief introduction to the core deep learning concepts used in our framework and some related work; Section 3 describes our novel deep learning framework for fault detection on time-series data; Section 4 outlines our experimental design and discusses the results of our experiments; and Section 5 will present our conclusions and some ideas for further work. Code to replicate all the experiments in this paper is available [13].

## 2. Background and Related Work

### 2.1. Convolutional Neural Networks

In the past decade, Convolutional Neural Networks (CNNs) have been responsible for major breakthroughs in computer vision and image processing [14,15,16], obtaining state-of-the-art results on a range on benchmark and real-world tasks. More recently, one-dimensional CNNs have shown significant promise in dealing with structured language data in tasks such as machine translation [17,18] and document classification [19,20]. In 2018, Bai et al. [21] showed that, for many sequence modeling tasks, 1D CNNs using current best practices such as dilated convolutions often perform as well as or better than recurrent neural network architectures.

Convolutional neural networks are a class of feed-forward neural networks consisting of multiple convolutional stages that perform the task of feature extraction and a single output stage that combines the extracted high-level features to predict the desired output. Figure 1 shows an example of a simple one-dimensional CNN architecture.

Each of the convolutional stages in Figure 1 shows a set of learnable convolutional filters followed by a pooling operation. These convolutional filters act to extract the high-level features (such as edges and curves in an image) from a supplied input by convolving a set of weights with the input and applying a non-linear activation function. The outputs of this are then fed into a pooling operation which reduces the spatial size of the features extracted by the convolutional filters whilst emphasizing the dominant features learned by each filter. As the input progresses through the convolutional stages (left to right on Figure 1), the network learns more problem-specific features.

### 2.2. Recurrent Neural Networks

Recurrent neural networks (RNNs) are capable of capturing the dynamic temporal behavior in an input sequence. RNNs do this by retaining and utilizing information about what has happened in previous time steps, allowing them to “remember” information about the entire input sequence. Whilst RNNs can theoretically capture input dependencies over long time intervals, Bengio et al. [22] have shown that, in practice, training such networks with gradient descent methods becomes less efficient as the temporal span of the input sequence dependencies increases—thus resulting in RNNs becoming difficult to train successfully.

The Long-Short Term Memory (LSTM) recurrent neural network architecture [23] (shown in Figure 2) overcomes some of the difficulties of training ordinary RNNs by introducing both a memory cell and a gating mechanism to improve the ability to learn from long input sequences. The memory cell provides a continuous path through the network, allowing multiple LSTM cells to be joined together without suffering from the “vanishing gradient problem” common to ordinary RNN architectures. Information is then added to this memory cell state via three gates:The forget gate, which controls how much of the previous state the LSTM should remember.The input gate, which controls how much new information the LSTM should add to its memory.The output gate, which controls how much of the memory we are going to use in the output of the state at the current time step.

The computation of the LSTM states at a given time step (*t*) can be expressed as [24]:(1)$$\begin{array}{cc} f_{t} & =\sigma (W_{f}x_{t}+U_{f}h_{t-1}+b_{f}) \end{array}$$(2)$$\begin{array}{cc} i_{t} & =\sigma (W_{i}x_{t}+U_{i}h_{t-1}+b_{i}) \end{array}$$(3)$$\begin{array}{cc} o_{t} & =\sigma (W_{o}x_{t}+U_{o}h_{t-1}+b_{o}) \end{array}$$(4)$$\begin{array}{cc} c_{t} & =f_{t}\circ c_{t-1}+i_{t}\circ \tanh\left(W_{c}x_{t}+U_{c}h_{t-1}+b_{c}\right) \end{array}$$(5)$$\begin{array}{cc} h_{t} & =o_{t}\circ \tanh\left(c_{t}\right) \end{array}$$
where:
$x_{t}$= the input vector$f_{t}$= the activation vector of the forget gate at time *t*$i_{t}$= the activation vector of input gate at time *t*$o_{t}$= the activation vector of the output gate at time *t*$c_{t}$= the memory cell activation vector at time *t*$h_{t}$= the hidden state activation vector at time *t*$W_{f}$, $W_{i}$, and $W_{o}$= the input kernels for the respective gates$U_{f}$, $U_{i}$, and $U_{o}$= the recurrent kernels for the respective gates$b_{f}$, $b_{i}$, $b_{o}$, and $b_{c}$= the biasesσ= the logistic sigmoid function∘= the matrix producttanh= the hyperbolic tangent activation function

Recently, Cho et al. [25] proposed the Gated Recurrent Unit (GRU) model as a variant of the LSTM architecture. The key difference is that the GRU architecture combines both the forget and input gates into a single “update gate”, producing a model that is simpler and has fewer weights to update and thus can train significantly faster.

Several researchers have compared GRU and LSTM based recurrent neural networks [26,27,28], with no clear winner emerging. A general rule-of-thumb is that GRU based RNNs train faster and seem to perform better for smaller amounts of data (as they are less prone to overfitting in such cases), but LSTM based RNNs have greater expressive power in capturing longer-term relationships in the data (but this comes at a cost of computational complexity and potential overfitting).

Whilst both LSTM and GRU models can capture input dependencies over long time intervals, they can sometimes struggle to identify dependencies if they are widely spaced (for example, if the first and last elements of a long sequence are important for making a correct prediction). Recently Bahdanau et al. [29] proposed an attention mechanism to help with this issue in neural machine translation. In their model of attention, a context vector (C) is used to capture information about the importance of the other elements in a given input sequence in making an accurate prediction.

### 2.3. Deep Learning in Intelligent Data Driven Fault Diagnosis

In the last few years, there has been increasing interest in applying deep learning-based approaches to data-driven fault diagnosis. These approaches fall broadly into two categories: those that reshape the input signals in some way into two dimensions, and those that operate on one-dimensional signals such as time-varying vibration data.

Guo et al. [30], Lu et al. [31] and Xia et al. [32] all reshaped the one-dimensional input signals into two-dimensional matrices of features and applied simple 2D CNN architectures to those feature matrices to categorize bearing damage. In contrast, Wen et al. [33] used a more sophisticated preprocessing method to turn the time domain signals into grayscale images and implemented a deeper 2D CNN (based on LeNet-5 [34]) to diagnose bearing faults across three different datasets.

Zhang et al. [35] proposed a deep one-dimensional CNN with a wide first layer kernel (WDCNN) capable of working directly on the raw temporal signals without the need for complex preprocessing. Zhuang et al. [36] also proposed a 1D CNN capable of operating on raw vibration signals but used dilated convolutions and residual connections to improve robustness to noise and domain shifts at the cost of some complexity in tuning the model parameters. Zhang et al. [37] adapted the WDCNN proposed in [35] to work with limited data—a common challenge in fault diagnosis where components are rarely allowed to run to failure.

## 3. The RNN-WDCNN Model for Fault Diagnosis

### 3.1. The RNN-WDCNN Model Architecture

In this paper, a novel dual-path deep learning network, combining a recurrent neural network path and a deep convolutional network path (RNN-WDCNN) is proposed and applied to fault diagnosis for rolling element bearings. This model architecture, shown in Figure 3, is inspired by the Long-Short Term Memory Fully Convolutional Network (LSTM-FCN) architecture proposed by Karim et al. [38]. LSTM-FCN has been shown to perform well across a range of benchmark tasks in time-series classification [39]; however, the small kernel size in the first convolutional layer hampers the ability of the fully convolutional network path to capture distant dependencies in the raw temporal signals from the noisy machine operating data (e.g., the raw vibration signals for bearing fault diagnosis used in this work). This problem is even more pronounced when these signals are sampled at a high sampling rate as the parts of interest in the signal are correspondingly further apart. As well as this, the relatively shallow nature of the convolutional pathway (using three convolutional stages) restricts the ability of the network to capture complex representations of low-level features.

To overcome these limitations, the concept of using a wide first layer kernel followed by multiple stages of convolutional layers with smaller kernels, proposed in Zhang et al. [35], is incorporated into the dual path model in place of the relatively shallow convolutional path. The deep convolutional path of the proposed model, therefore, consists of five convolutional stages for feature extraction and a final dimension reduction stage to compress the learned feature representations. Each of the convolutional stages comprises a set of 1D convolutional filters to learn the features of significance from the raw input signals, a rectified linear unit activation function (ReLU) to introduce non-linearities into the network, a batch normalization block [40] to improve the training process by reducing the internal co-variance shift by normalizing layer inputs between batches, and a max-pooling block to down-sample the input representation and improve the local translation invariance (i.e., the robust of the learned feature map to the position of the features within the input signals). The first of these convolutional stages in this deep convolutional path uses a very wide filter kernel both to suppress high-frequency noise in the input signals and to capture distant dependencies, whilst the remaining convolutional stages use smaller kernel sizes and serve to allow the convolutional path to acquire complex representations of the features of the input signals.

As can be seen from the model architecture in Figure 3, as well as using a very wide convolutional filter kernel in the first layer of the convolutional path, a 1D convolutional layer with a wide filter kernel is also added before the recurrent neural network block in the RNN path. This helps primarily to suppress high-frequency noise in the raw input signals but also to learn useful features to feed into the RNN block. The primary job of the RNN block in the dual-path architecture is to help capture dynamic temporal features that span large numbers of time steps. These temporally diverse features learned by the RNN path act to complement the more locally situated features learned by the convolutional path and improve the final classification results. As will be discussed in Section 3.2, several different candidate RNN architectures are investigated in this work, including those augmented with an attention mechanism (as described in [29]).

Finally, the deep convolutional path and the RNN path are concatenated together and fed to a output classification layer which uses the softmax function to transform the raw logits into a probability distribution corresponding to the classes of bearing fault (where all elements lie in the range [0, 1] and sum to 1). This softmax function is defined as:(6)$$Softmax\left(x_{i}\right)=\frac{exp\left(x_{i}\right)}{\sum_{j}exp\left(x_{j}\right)},$$
where $x_{i}$ and $x_{j}$ are the *i*-th and *j*-th elements of the raw logits vector from the model, respectively.

The parameters of the two pathways are outlined in Table 1. As shown in Table 1, the RNN block in the recurrent pathway is comprised of 128 RNN cells as this strikes a good balance between the ability to capture long-term dynamics in the signal and the complexity of the recurrent path. This value was found by an initial hyper-parameter search.

### 3.2. Training the RNN-WDCNN Model

Xavier uniform initialization [41] was used to initialize all the weights of the RNN-WDCNN model (for both the RNN and WDCNN paths). To update the model weights during training, gradient descent-based backpropagation was applied using the Stochastic Gradient Descent (SGD) optimizer and a Cyclical Learning Rate (CLR) strategy [42], with the categorical cross-entropy function used to evaluate the training and validation loss of the network. Categorical cross-entropy compares the distribution of the output predictions of the model with the one-hot encoded ground truth vector. The closer the model’s outputs are to this vector, the lower the resulting loss. In this work, the ground truth vector represents the one-shot encoded class of the rolling element bearing fault indicated by the input vibration signals.

The range of the learning rates for the CLR strategy was chosen by gradually increasing the learning rate from very low ($1\times 10^{-10}$) to very high (10) and plotting the behavior of the loss value. The maximum and minimum (i.e., base) learning rates were then chosen as the points between when the loss decreases most sharply, as shown in Figure 4. Learning rates were adjusted between the base learning rate and the maximum learning rate using the triangular2 policy (whereby the maximum learning rate is halved after each cycle), with the step size chosen so that there would be three cycles over the course of the training process (with the training process finishing at the end of the third cycle). A batch size of 8 was used during this training process as small batch sizes have been shown to improve the generalization performance of a trained model by helping the optimizer converge to flatter minima in the search landscape [43].

Recurrent dropout [44] and standard dropout [45] were applied during training to reduce over-fitting and improve model generalization. Recurrent dropout was applied with a probability of 0.1 to both the inputs and hidden states of the RNN cells, and standard dropout was applied to both convolutional and RNN paths (with a probability of 0.5) before concatenation. A variety of recurrent neural network architectures have been investigated in this work and their performance is assessed in Section 4. These include:A basic LSTM model (LSTM)A basic GRU model (GRU)An LSTM model with Attention (ALSTM)A GRU model with Attention (AGRU)

## 4. Experimental Results

### 4.1. Data Description

To validate the effectiveness of the novel dual-path recurrent neural network with wide first kernel and deep convolutional pathway proposed in this work for intelligent data-driven fault diagnosis applications, the benchmark bearing fault dataset from Case Western Reserve University (CWRU) Bearing Data Center [46] is used. The CWRU bearing fault dataset has been widely used in the literature for investigating both conventional fault diagnosis techniques [47,48] and data-driven fault diagnosis methods using machine learning [49,50] and deep learning [35,36,37]. The data used in this study were collected from both the drive end accelerometer (close to the fault location) and the fan end accelerometer (remote from the fault location) of the test apparatus shown in Figure 5. Data were collected over a range of motor loads (from 0 hp to 3 hp) and for a variety of single point damage conditions, with bearing damage introduced by using electro-discharge machining (EDM) to induce faults of varying types and severity (see Table 2 for a list of fault conditions).

Contrary to much of the literature, in this work, we use the 48 kHz sampled vibration signals from the fan end and drive end accelerometers rather than the 12 kHz sampled data. Smith and Randall [48] have stated that, although for this dataset the 12 kHz sampled data may be easier to diagnose, many bearing faults manifest themselves at high frequencies and thus an effective fault diagnosis framework must be capable of operating on data sampled at a high sampling rate.

One drawback of deep learning models is their need for large amounts of data for training the model; however, in fault diagnosis applications it is rare to have large amounts of data due to the inherent difficulties in running machinery in fault conditions for prolonged periods of time. In this study, data augmentation was used to increase the amount of training data available to train the model and help reduce over-fitting. To do this, a sliding window approach was used to extract multiple overlapping samples from the input sequences (as shown in Figure 6). Each input sequence was acquired under a single fault condition, so the augmented sequences were all assigned the same fault label as the original input sequence. The experiments in this study use a window length of 4096 data points and a step size of 64 to create the training data. A window of 4096 data points was chosen to capture at least two complete revolutions of the load-bearing shaft—i.e., the rotating motor shaft—and therefore impacts on the bearing fault area. The number of data points in a complete revolution of the load-bearing shaft is given by $N=\frac{F_{s}\times 60}{\omega }$ (where *N* is the number of data points, $F_{s}$ is the sampling frequency, and ω is the rotating shaft speed in rpm). As Table 3 shows, the rotating shaft speed varies between 1772 rpm and 1730 rpm for the operating conditions investigated in this work. At a sampling rate of 48 kHz, this gives between 1626 and 1665 data points per complete revolution which means a window length of 4096 data points is guaranteed to capture at least two bearing fault impacts. This approach is similar to that taken by other works in the literature that use the 12 kHz sampled data [35,36]. The test data was created using the same window length but without samples overlapping. The labels for both the training and testing datasets were evenly distributed between the fault classes (as shown in Table 4) for each of the load conditions.

For the experiments outlined in the rest of this section, the proposed RNN-WDCNN model is compared to the WDCNN model proposed in Zhang et al. [35] and the state-of-the-art deep learning-based models (SRDCNN) model proposed in Zhuang et al. [36] (see Section 2.3 for more discussion of these approaches). These models have been re-implemented based on the descriptions in the respective papers to allow for an equitable comparison of the performance of all three models on the 48 kHz sampled data. All models have been implemented in the Keras deep learning framework [51] with the TensorFlow back end [52] and code is available at https://github.com/al3xsh/rolling-element-bearing-fault-detection. The three deep learning-based models are further compared to two conventional machine learning approaches using Fast Fourier Transform (FFT)-based features (one using a multilayer perceptron (FFT-MLP), and one using support vector machines (FFT-SVM)). The results presented in the following sections are the mean and standard deviation for 10 independent runs of each model, with the 10 runs performed using stratified 10-fold cross-validation (so for each run the model is trained with a slightly different subset of the training data—maintaining the balance of classes in the dataset—and then the performance validated on the held out part of the training set, before finally evaluating the model under different working conditions or environmental noise levels).

The accuracy of the fault classification is considered as the primary performance metric (because the fault classes are evenly balanced and the importance of detecting each fault class is equal); however, results for precision, recall, and the F1 score are also presented to allow greater insight into the performance of all the models tested in this section. Precision provides information about the ability of a classifier to correctly label samples, whilst recall shows the ability of a classifier to find all samples that belong to a class. The precision metric thus helps to understand the false positives generated by a model and recall helps to understand the false negatives. The F1 score is a weighted mean of both precision and recall. Note that, because multiple fault classes are considered, the precision, recall, and F1 score metrics are macro-averaged (i.e., each metric is calculated per class and then the macro-average is calculated by taking the arithmetic mean over all the classes).

### 4.2. Performance under Different Working Loads

In this set of experiments, the domain adaptation performance of the RNN-WDCNN model is investigated to see how well it generalizes to similar data acquired under different operating conditions (i.e., motor load and shaft speed). datasets using three distinct sets of operating conditions were used in this study. These are shown in Table 3.

In this section two scenarios are considered:The model is trained using data acquired under one set of operating conditions and then tested individually against unseen data acquired under different, but fixed, operating conditions.The model is trained using data from two different sets of operating conditions and then tested against unseen data acquired under the third set of operating conditions (again fixed, but different from the two used in training).

#### 4.2.1. Scenario 1: Single Source Domain to a Single Target Domain

Table 5 provides details of the arrangement of the datasets used in this scenario. As mentioned in Section 4.1, the data used for training has been augmented using a sliding window approach, and the data used for testing (the target domain in Table 5) is split into non-overlapping windows of the same length as the training samples. Figure 7 shows the differences in performance in this scenario between the four versions of RNN-WDCNN considered in this work (i.e., models using the GRU, LSTM, AGRU, and ALSTM based recurrent pathways).

It can be seen from Figure 7 that there are only small differences in the domain adaptation performance between the different recurrent architectures investigated for use with RNN-WDCNN, with the basic LSTM model (LSTM-WDCNN) performing slightly better on average across the domain shifts considered in this scenario. It is only in the domain shift from dataset C to dataset B where a different recurrent architecture (the LSTM pathway with attention—ALSTM-WDCNN) performs significantly better. For the rest of the experiments in this section, we therefore, use the LSTM-based recurrent pathway in the RNN-WDCNN framework.

Figure 8 shows a comparison between the accuracy of the proposed RNN-WDCNN model, two state-of-the-art deep learning-based models (WDCNN and SRDCNN), and two FFT based methods (FFT-MLP and FFT-SVM) in this scenario. Table 6, Table 7 and Table 8 also present performance data for the precision, recall, and F1 score metrics (respectively) for this domain adaptation scenario. Figure 8 and Table 6, Table 7 and Table 8 show that all three deep learning-based models generally perform well in adapting to small changes in operating conditions (i.e., when the source domain and target domain are reasonably similar); however, as the operating conditions of the two domains become further apart the proposed RNN-WDCNN model can be seen to outperform the other models tested. It can also be seen from Figure 8 and Table 6, Table 7 and Table 8 that the FFT based models generally struggle to adapt to different working domains, with their overall performance significantly worse than the deep learning-based models.

Figure 9 shows the confusion plots for the proposed RNN-WDCNN model when tested under different operating characteristics from those on which it was trained. It can be seen from these results that the key problematic dataset for domain adaptation is dataset A—with the RNN-WDCNN model frequently mistaking minor rolling element damage in dataset A (fault B007) for moderate outer raceway damage (fault OR014) when trained with either dataset B or C. However, Smith and Randall [48] have shown that many of the rolling element faults in the Case Western Reserve University Bearing Fault data also exhibit signs of inner and outer raceway faults when analyzed using conventional methods, with the vibration signatures of these other faults often overwhelming the rolling element fault vibration signature. The combination of a shift in operating conditions and the presence of an outer raceway fault, as well as the minor rolling element damage comprising the target fault condition, could explain the classification of B007 as OR014.

#### 4.2.2. Scenario 2: Multiple Source Domains to a Single Target Domain

As can be seen from the results presented in Section 4.2.1, although the proposed RNN-WDCNN model performs extremely well when the operating conditions of the source and target domains are similar, the performance suffers somewhat when the operating conditions of the two domains are further apart. In particular, as shown in the confusion plots in Figure 9, this domain shift can act to hinder classification when there are potential additional undiagnosed fault conditions. Following the approach in Zhang et al. [37], this scenario considers the case where labeled fault data from multiple operating conditions is available. In this case, the model can be trained using data acquired under these multiple operating conditions to provide a more robust fault diagnosis system. Table 9 shows the arrangement of the data in the scenario considered in this section, with Figure 10 and Table 10, Table 11 and Table 12 showing the results of these experiments.

As shown in Figure 10 and Table 10, Table 11 and Table 12, the use of training data from multiple operating conditions significantly improves the performance of all the algorithms tested in this scenario in terms of accuracy, precision, recall, and F1 score. This combination of multiple source domains for training has the greatest impact on the success of identifying fault conditions in the unseen dataset A, where all algorithms now achieve greater than 90% accuracy and four of the tested architectures achieve greater than 95% accuracy, with the FFT based models performing particularly well in this domain adaptation case. Overall in this scenario, RNN-WDCNN and WDCNN perform best of all the algorithms tested (with no significant difference between the overall performance of the two algorithms).

Figure 11 shows two of the confusion plots for the proposed RNN-WDCNN model in this domain adaptation scenario (the case of unseen dataset B is omitted, since RNN-WDCNN obtains 100% accuracy under these conditions). It can be seen that, although the use of data from multiple operating conditions significantly improves the classification performance (particularly with respect to the classification of severe rolling elements faults—fault B021), in the more difficult case (training on datasets B and C and testing on unseen dataset A) RNN-WDCNN still often mistakes minor rolling element damage in dataset A (fault B007) for moderate outer raceway damage (fault OR014). This lends weight to the hypothesis that the data for minor rolling element damage under 1 hp load in the CWRU dataset also contains an undiagnosed outer raceway fault (as discussed in Section 4.2.1, above).

### 4.3. Performance under Varying Amounts of Noise

In this section, the robustness of the proposed RNN-WDCNN model to noisy environments is investigated. Under real-world industrial operation, there are often many sources of electronic noise (such as complex wiring runs, brushed DC motors, and high speed switching transients), and so an effective data-driven intelligent fault diagnosis platform must be able to operate accurately under these noisy conditions.

In this scenario, 50% of the data for each set of operating conditions is held back and used to create an unseen test set. The models are then trained on the remaining 50% of the data (with sliding window data augmentation used to artificially increase the number of samples available for training). Once the model is trained, it is then tested under noisy conditions using the data from the test set with varying amounts of Gaussian white noise added to the signals to obtain signals with different signal-to-noise ratios (from −4 dB to 10 dB). The signal-to-noise ratio (SNR) is defined as:(7)$$SNR_{dB}=10\log_{10}\left(\frac{P_{signal}}{P_{noise}}\right),$$
where $P_{signal}$ and $P_{noise}$ are the power of the signal and the power of the noise, respectively.

As in the domain adaptation experiments in Section 4.2, firstly the performance of the different recurrent pathways in the presence of noise is investigated. Figure 12 shows that the performance is again similar; however, the LSTM model performs slightly better than the other architectures in the presence of larger amounts of noise. The LSTM based recurrent pathway is therefore used in the RNN-WDCNN framework for the rest of the experiments in this section.

Figure 13 and Table 13, Table 14 and Table 15 show a comparison of the noise rejection performance between the proposed RNN-WDCNN model, the two state-of-the-art deep learning-based models (WDCNN and SRDCNN), and the two FFT based methods (FFT-MLP and FFT-SVM). It can clearly be seen that, under small amounts of noise, all the algorithms perform well (with better than 90% accuracy when the signal-to-noise ratio is 4 dB or greater). However, as the noise level increases, RNN-WDCNN significantly outperforms the other algorithms—achieving better than 98% accuracy at a signal-to-noise ratio of −4 dB.

Figure 14 shows the confusion plot for the proposed RNN-WDCNN model under the most severe noise conditions considered in this experiment (i.e., when the SNR is equal to −4 dB). It can be seen from Figure 14 that the largest effect of the introduction of noise on the classification accuracy of RNN-WDCNN is in the consideration of rolling element faults. This may be partly because, as Smith and Randall [48] have shown (and as discussed in Section 4.2.1 above), many of the rolling element faults in the CWRU Bearing Fault data also exhibit signs of inner and outer raceway faults and the addition of noise may act to obscure the true fault condition. However, even under this severe noise condition, the classification accuracy of RNN-WDCNN is excellent (with greater than 98% accuracy).

### 4.4. Inference and Training Time Costs

A key consideration of any fault diagnosis method is the time cost required to produce a classification result. In this section, we consider both the time needed to train each of the models considered in Section 4.2 and Section 4.3 and the inference time of each model (i.e., the time needed to classify a single sequence using a trained model). Table 16 shows the average training time (in seconds) for each model on each of the three scenarios considered in Section 4.2.1, Section 4.2.2 and Section 4.3, and the average inference time (in milliseconds) for a single 4096 sample sequence. The training times presented in this section are averaged over 10 independent runs for each algorithm. The six deep learning algorithms considered in this paper are trained for 25 epochs using the cyclical learning rate strategy described in Section 3.2 (which experiments showed resulted in all six models converging). The two FFT-based methods were trained until convergence by monitoring the improvement in loss value and stopping once no further improvement could be seen. The inference time for each model was measured whilst processing a single sample at a time and averaged over 100,000 vibration sequences. This inference time is the same for each of the different scenarios considered in this paper because the sequence length remains the same. Note that, for the FFT based methods, this inference time includes the time taken to extract the FFT features. All models were trained and executed on a PC with Intel i7-7700 CPU, 32 GB of RAM, and nVidia GeForce GTX 1070 GPU.

As shown by Table 16, although the deep learning methods generally take much longer to train than the FFT-based machine learning methods in each of the scenarios considered, the inference times of the deep learning methods are significantly shorter. This is because the deep learning methods operate directly on the raw temporal vibration signals rather than requiring a time-consuming feature extraction step. The fastest method in terms of inference time is WDCNN (taking 1.56 ms to process a single vibration sequence) but the proposed RNN-WDCNN algorithm without attention still performs well (taking 3.53 ms to process a single vibration sequence using the LSTM variant). In the real-world deployment of an intelligent fault diagnosis system, it is the inference time of the algorithm that is the most important timing metric. This is because the training of the model is a one time cost and can be done offline, whereas the classification of a sequence must be done continuously and in an online fashion. In the dataset considered in this work, each 4096 element vibration sequence represents 85 ms of data (with the data acquired at 48 kHz). With an inference time of 3.53 ms, this would allow the proposed RNN-WDCNN intelligent fault diagnosis system to process the vibration data online in real-time.

## 5. Conclusions and Further Work

The key contribution of this work is the development of an intelligent fault diagnosis framework capable of operating in real-time on unprocessed sensor signals to allow early detection of impending failures under variable load conditions. The proposed novel dual-path deep learning model obtains state-of-the-art classification performance, while also addressing many of the problems with existing intelligent data-driven fault diagnosis techniques such as being able to operate on raw time series data (with no need for complex feature engineering) and robustness to noise and changes in operating conditions. The novel dual-path model proposed in this work outperforms current-state-of-the-art methods both using data from shifted domains (i.e., when the testing data is acquired under different operating conditions from those under which the model was trained) and using data acquired under noisy conditions. The proposed model also performs well in terms of inference time, being able to classify input sequences faster than conventional FFT based methods.

The experiments conducted in this paper on the Case Western Reserve University Bearing Center dataset have shown that our proposed novel dual-path model outperforms several conventional classifiers and current state-of-the-art algorithms. Surprisingly, the inclusion of attention in the recurrent pathway of the RNN-WDCNN model did not make a significant difference to the performance of our model in either shifted domains or under noisy conditions. This may be because, although the attention mechanism helps to capture widely spaced dependencies, these widely spaced features do not transfer well between domains. Further work is planned to perform more detailed studies into the effects of the individual elements of our model, including looking at the performance of the model in different applications where domain shifts are less common.

Although the Case Western Reserve University Bearing Center dataset used in this study has become the de facto benchmark for vibration-based bearing fault diagnosis, Smith and Randall [48] have shown that it has several flaws (including undiagnosed fault conditions that can significantly affect classification performance). In further work, we would like to test our fault diagnosis framework on a wider range of data—including real-world fault conditions (rather than just artificial faults induced using EDM). Another promising area for further work is the diagnosis of motor faults from other signals (such as motor current signals or torque data) rather than vibration data as these can often be acquired directly from the programmable logic controller (PLC) rather than requiring the installation of additional sensors.

## Figures and Tables

**Figure 1 sensors-20-05112-f001:**
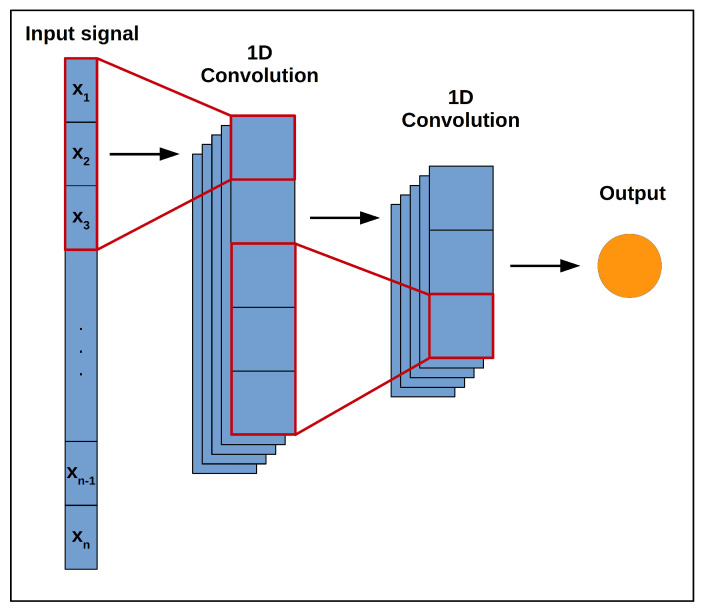
Simple 1D convolutional neural network (CNN) architecture with two convolutional layers.

**Figure 2 sensors-20-05112-f002:**
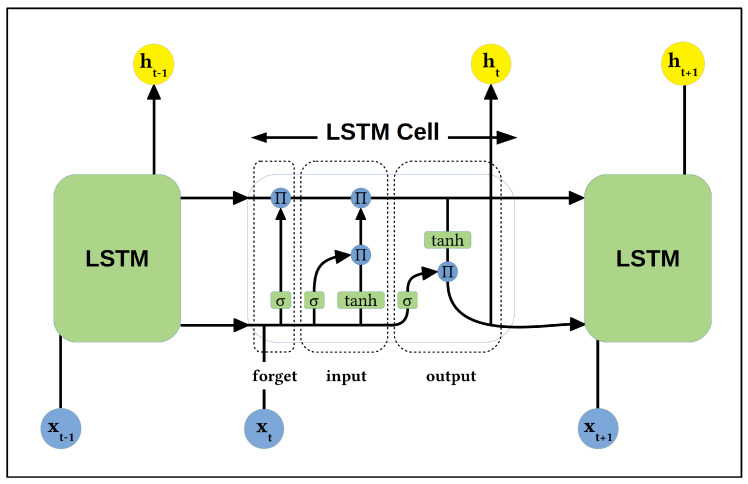
Basic Long-Short Term Memory (LSTM) architecture.

**Figure 3 sensors-20-05112-f003:**
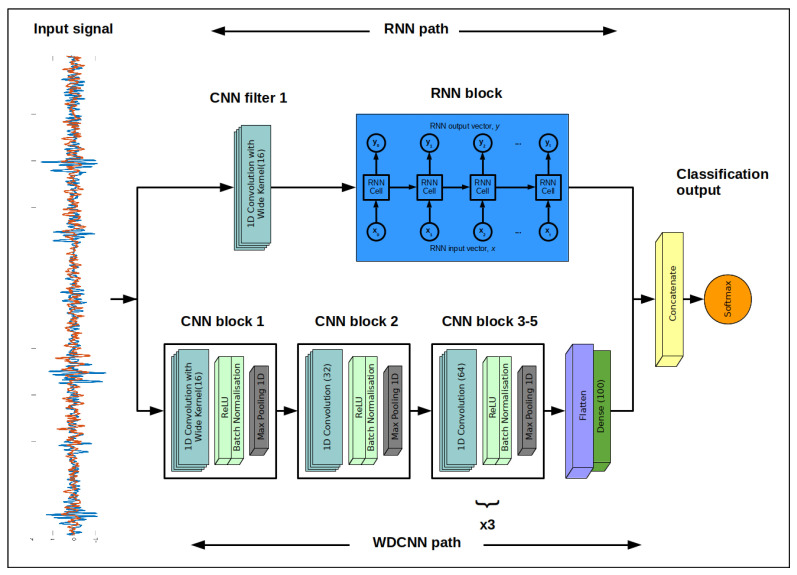
The proposed recurrent neural network with wide first kernel and deep convolutional path (RNN-WDCNN).

**Figure 4 sensors-20-05112-f004:**
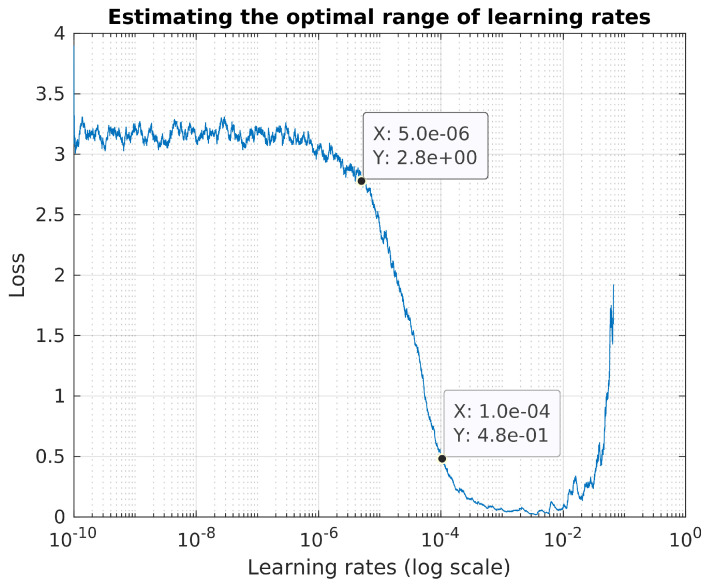
Finding the optimal learning rate range to use in the cyclical learning rate strategy.

**Figure 5 sensors-20-05112-f005:**
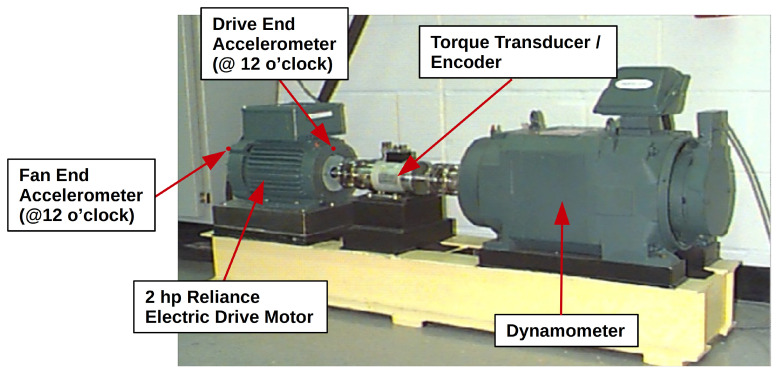
The Case Western Reserve University Bearing Center test apparatus (original image taken from [46]).

**Figure 6 sensors-20-05112-f006:**
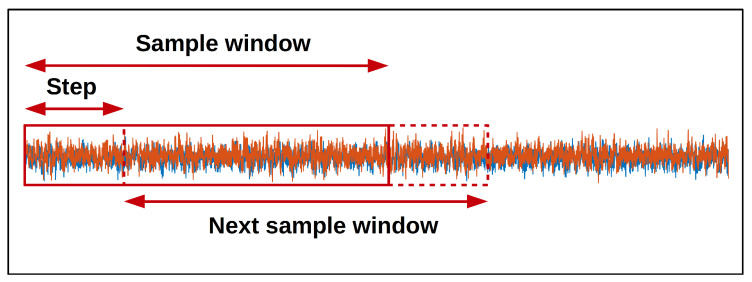
Sliding window data augmentation.

**Figure 7 sensors-20-05112-f007:**
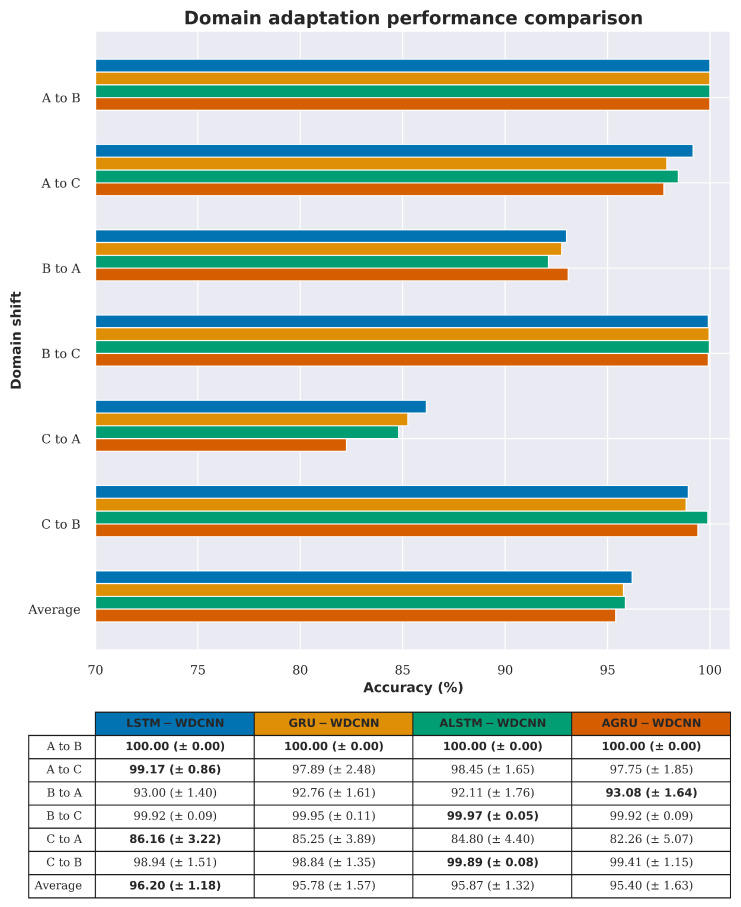
Domain adaptation accuracy comparison between the different recurrent pathways in recurrent neural network with a wide first kernel and deep convolutional neural network pathway (RNN-WDCNN) for scenario 1 (best values are highlighted in bold).

**Figure 8 sensors-20-05112-f008:**
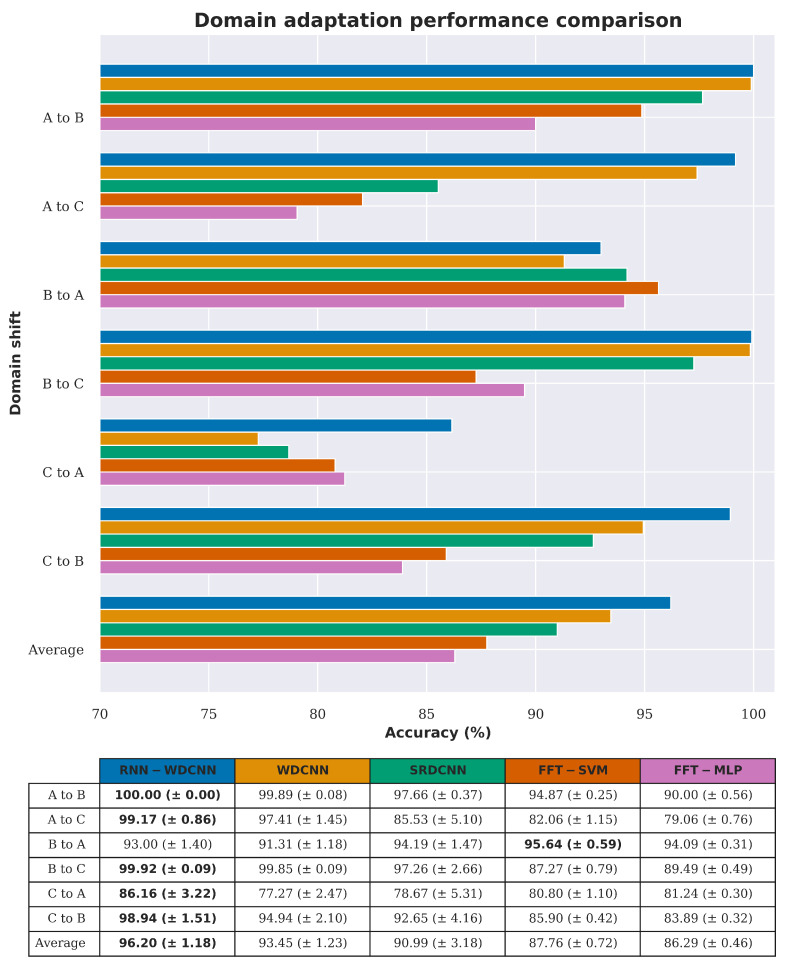
Domain adaptation accuracy comparison between RNN-WDCNN and WDCNN, state-of-the-art deep learning-based models (SRDCNN), and FFT-based methods for scenario 1 (best values are highlighted in bold).

**Figure 9 sensors-20-05112-f009:**
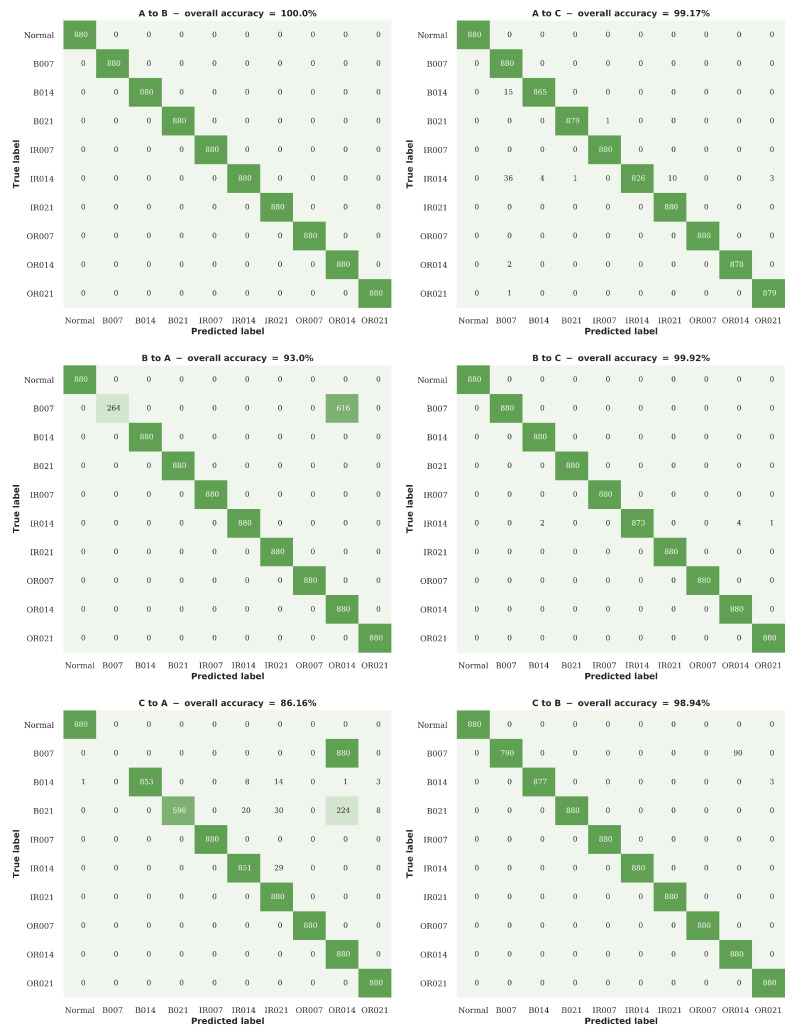
Confusion plots for the RNN-WDCNN model on the 6 different domain adaptation cases considered in scenario 1.

**Figure 10 sensors-20-05112-f010:**
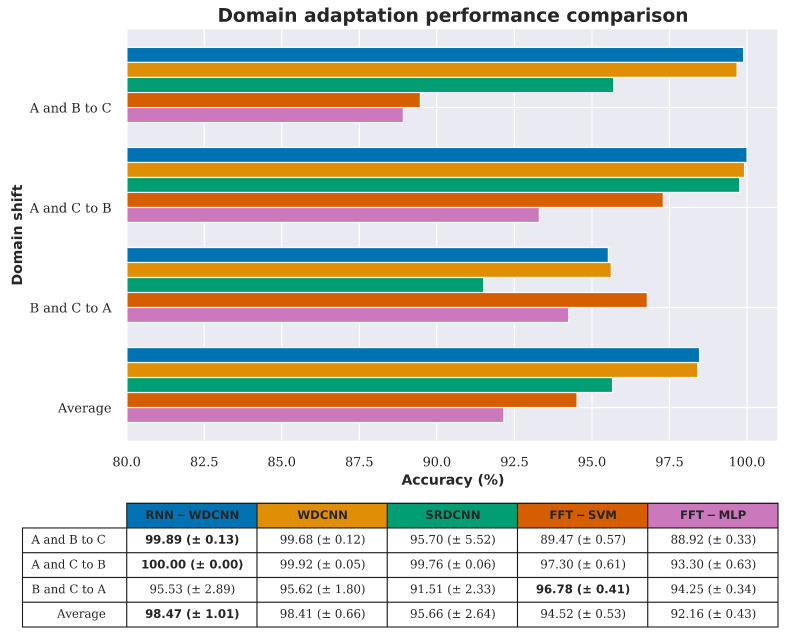
Domain adaptation accuracy comparison between RNN-WDCNN and WDCNN, SRDCNN, and FFT-based methods for scenario 2 (best values are highlighted in bold).

**Figure 11 sensors-20-05112-f011:**
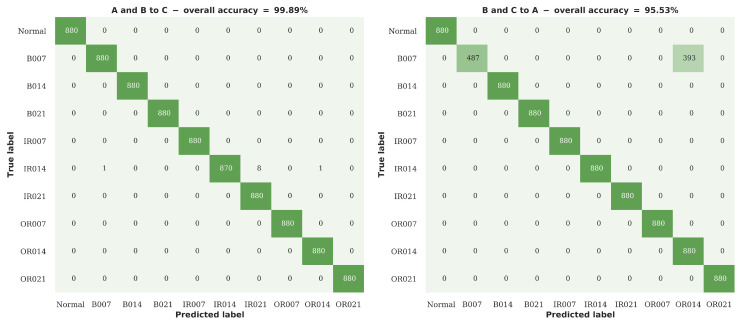
Confusion plots for the RNN-WDCNN model on the two of the domain adaptation cases considered in scenario 2.

**Figure 12 sensors-20-05112-f012:**
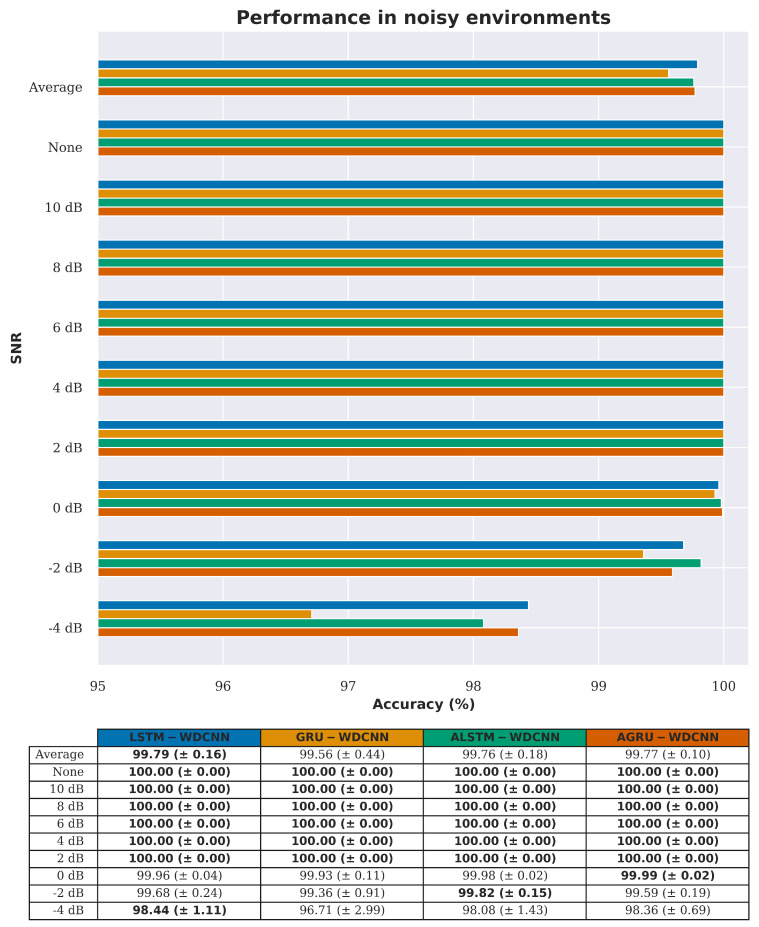
Noise rejection accuracy comparison between the different recurrent pathways in RNN-WDCNN (best values are highlighted in bold).

**Figure 13 sensors-20-05112-f013:**
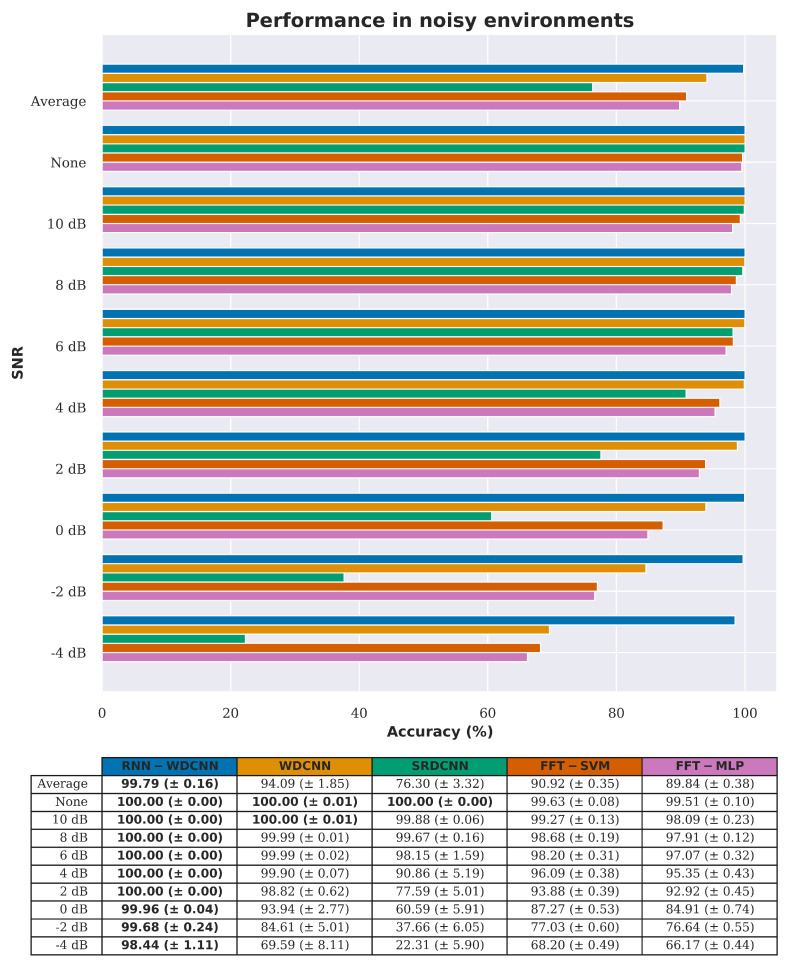
Noise rejection accuracy comparison between RNN-WDCNN and WDCNN, SRDCNN, and FFT based methods (best values are highlighted in bold).

**Figure 14 sensors-20-05112-f014:**
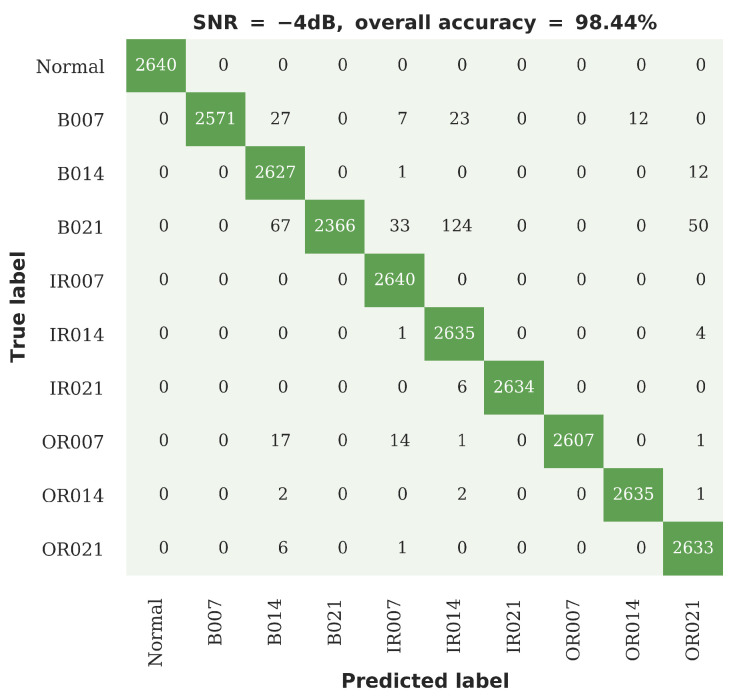
Confusion plot for the RNN-WDCNN model in the case of severe additive noise (SNR −4 dB).

**Table 1 sensors-20-05112-t001:** Details of the proposed RNN-WDCNN model structure.

**Convolutional Pathway**				
**No.**	**Layer Type**	**Kernel Size**	**Stride**	**Number of Kernels**
C1	1D Convolution	256 × 1	2 × 1	16
C2	Max Pooling 1D	2 × 1	2 × 1	16
C3	1D Convolution	3 × 1	2 × 1	32
C4	Max Pooling 1D	2 × 1	2 × 1	32
C5	1D Convolution	3 × 1	2 × 1	64
C6	Max Pooling 1D	2 × 1	2 × 1	64
C7	1D Convolution	3 × 1	2 × 1	64
C8	Max Pooling 1D	2 × 1	2 × 1	64
C9	1D Convolution	3 × 1	2 × 1	64
C10	Max Pooling 1D	2 × 1	2 × 1	64
C11	Fully connected	100	N/A	1
**Recurrent Pathway**				
**No.**	**Layer Type**	**Kernel Size**	**Stride**	**Number of Kernels**
R1	1D Convolution	256 × 1	2 × 1	16
R2	Max Pooling 1D	2 × 1	2 × 1	16
R3	RNN block	N/A	N/A	128
**Output**				
**No.**	**Layer Type**	**Kernel Size**	**Stride**	**Number of Kernels**
O1	Softmax	10	N/A	1

**Table 2 sensors-20-05112-t002:** Bearing fault descriptions for the Case Western Reserve University (CWRU) dataset.

Fault ID	Fault Cause	Severity
1	Normal	N/A
2	Inner Race Fault	0.007 inch
3	Ball Fault	0.007 inch
4	Outer Race Fault	0.007 inch
5	Inner Race Fault	0.014 inch
6	Ball Fault	0.014 inch
7	Outer Race Fault	0.014 inch
8	Inner Race Fault	0.021 inch
9	Ball Fault	0.021 inch
10	Outer Race Fault	0.021 inch

**Table 3 sensors-20-05112-t003:** Operating conditions considered in these experiments.

Dataset	Motor Load	Shaft Speed
A	1 hp	1772 rpm
B	2 hp	1750 rpm
C	3 hp	1730 rpm

**Table 4 sensors-20-05112-t004:** Fault label distribution for each load condition (note that each load condition consists of the same number of training and testing samples).

Fault ID	Total Training Sample (with Data Augmentation)	Total Testing Samples
1	5625	88
2	5625	88
3	5625	88
4	5625	88
5	5625	88
6	5625	88
7	5625	88
8	5625	88
9	5625	88
10	5625	88

**Table 5 sensors-20-05112-t005:** Arrangement of the data for domain adaptation scenario 1.

Training Domain (Labeled Data)	Target Domains (Unlabeled Data)
Dataset A	Dataset B & Dataset C
Dataset B	Dataset A & Dataset C
Dataset C	Dataset A & Dataset B

**Table 6 sensors-20-05112-t006:** Precision scores of the models considered in this work (as a percentage) for the domain adaptation task considered in scenario 1 (best values are highlighted in bold).

Domain Shift	RNN-WDCNN	WDCNN	SRDCNN	FFT-SVM	FFT-MLP
**A to B**	**100.00 (± 0.00)**	99.89 (± 0.07)	97.89 (± 0.34)	94.97 (± 0.24)	90.07 (± 0.60)
**A to C**	**99.25 (± 0.72)**	97.86 (± 1.01)	88.66 (± 4.66)	81.90 (± 1.02)	79.29 (± 0.63)
**B to A**	95.92 (± 0.48)	95.34 (± 0.33)	**96.32 (± 0.65)**	95.63 (± 0.62)	94.69 (± 0.24)
**B to C**	**99.92 (± 0.09)**	99.85 (± 0.09)	97.71 (± 1.95)	87.59 (± 0.73)	89.68 (± 0.44)
**C to A**	**83.46** (± 1.29)	80.41 (± 1.77)	77.03 (± 6.51)	75.86 (± 1.24)	78.95 (± 0.64)
**C to B**	**99.17 (± 1.09)**	96.24 (± 1.06)	93.17 (± 6.03)	85.27 (± 0.61)	83.12 (± 0.49)
**Average**	**96.29 (± 0.61)**	94.93 (± 0.72)	91.80 (± 3.36)	86.87 (± 0.74)	85.97 (± 0.51)

**Table 7 sensors-20-05112-t007:** Recall scores of the models considered in this work (as a percentage) for the domain adaptation task considered in scenario 1 (best values are highlighted in bold).

Domain Shift	RNN-WDCNN	WDCNN	SRDCNN	FFT-SVM	FFT-MLP
**A to B**	**100.00 (± 0.00)**	99.89 (± 0.08)	97.66 (± 0.37)	94.88 (± 0.25)	90.00 (± 0.56)
**A to C**	**99.17 (± 0.86)**	97.41 (± 1.45)	85.53 (± 5.10)	82.06 (± 1.15)	79.06 (± 0.76)
**B to A**	93.00 (± 1.40)	91.31 (± 1.18)	94.19 (± 1.47)	**95.64 (± 0.59)**	94.09 (± 0.31)
**B to C**	**99.92 (± 0.09)**	99.85 (± 0.09)	97.26 (± 2.66)	87.27 (± 0.79)	89.49 (± 0.49)
**C to A**	**86.16 (± 3.22)**	77.27 (± 2.47)	78.67 (± 5.31)	80.80 (± 1.10)	81.24 (± 0.30)
**C to B**	**98.94 (± 1.51)**	94.94 (± 2.10)	92.65 (± 4.16)	85.90 (± 0.42)	83.89 (± 0.32)
**Average**	**96.20 (± 1.18)**	93.45 (± 1.23)	90.99 (± 3.18)	87.76 (± 0.72)	86.29 (± 0.46)

**Table 8 sensors-20-05112-t008:** F1 scores of the models considered in this work (as a percentage) for the domain adaptation task considered in scenario 1 (best values are highlighted in bold).

Domain Shift	RNN-WDCNN	WDCNN	SRDCNN	FFT-SVM	FFT-MLP
**A to B**	**100.00 (± 0.00)**	99.89 (± 0.08)	97.64 (± 0.38)	94.86 (± 0.25)	89.98 (± 0.55)
**A to C**	**99.16 (± 0.88)**	97.35 (± 1.53)	83.92 (± 5.92)	81.56 (± 1.12)	78.68 (± 0.83)
**B to A**	91.87 (± 2.12)	89.51 (± 1.75)	93.56 (± 1.95)	**95.60 (± 0.61)**	94.14 (± 0.32)
**B to C**	**99.92 (± 0.09)**	99.85 (± 0.09)	97.18 (± 2.79)	86.91 (± 0.82)	89.16 (± 0.50)
**C to A**	**82.97 (± 3.28)**	72.74 (± 3.09)	74.40 (± 6.79)	77.42 (± 1.07)	78.66 (± 0.31)
**C to B**	**98.91 (± 1.57)**	94.75 (± 2.42)	90.89 (± 5.69)	82.55 (± 0.38)	82.19 (± 0.38)
**Average**	**95.47 (± 1.32)**	92.35 (± 1.49)	89.60 (± 3.92)	86.48 (± 0.71)	85.47 (± 0.48)

**Table 9 sensors-20-05112-t009:** Arrangement of the data for domain adaptation scenario 2.

Training Domains (Labeled Data)	Target Domain (Unlabeled Data)
Dataset A & Dataset B	Dataset C
Dataset A & Dataset C	Dataset B
Dataset B & Dataset C	Dataset A

**Table 10 sensors-20-05112-t010:** Precision scores of the models considered in this work (as a percentage) for the domain adaptation task considered in scenario 2 (best values are highlighted in bold).

Domain Shift	RNN-WDCNN	WDCNN	SRDCNN	FFT-SVM	FFT-MLP
**A and B to C**	**99.89 (± 0.13)**	99.69 (± 0.12)	96.95 (± 3.46)	89.54 (± 0.59)	88.84 (± 0.31)
**A and C to B**	**100.00 (± 0.00)**	99.92 (± 0.05)	99.77 (± 0.06)	97.39 (± 0.54)	93.52 (± 0.58)
**B and C to A**	**97.18 (± 1.48)**	97.05 (± 0.84)	91.44 (± 5.64)	96.92 (± 0.38)	94.71 (± 0.35)
**Average**	**99.02 (± 0.54)**	98.89 (± 0.34)	96.05 (± 3.05)	94.61 (± 0.51)	92.35 (± 0.41)

**Table 11 sensors-20-05112-t011:** Recall scores of the models considered in this work (as a percentage) for the domain adaptation task considered in scenario 2 (best values are highlighted in bold).

Domain Shift	RNN-WDCNN	WDCNN	SRDCNN	FFT-SVM	FFT-MLP
**A and B to C**	**99.89 (± 0.13)**	99.68 (± 0.12)	95.70 (± 5.52)	89.47 (± 0.57)	88.92 (± 0.33)
**A and C to B**	**100.00 (± 0.00)**	99.92 (± 0.05)	99.76 (± 0.06)	97.30 (± 0.61)	93.30 (± 0.63)
**B and C to A**	95.53 (± 2.89)	95.62 (± 1.80)	91.51 (± 2.33)	**96.78 (± 0.41)**	94.25 (± 0.34)
**Average**	**98.47 (± 1.01)**	98.41 (± 0.66)	95.66 (± 2.64)	94.52 (± 0.53)	92.16 (± 0.43)

**Table 12 sensors-20-05112-t012:** F1 scores of the models considered in this work (as a percentage) for the domain adaptation task considered in scenario 2 (best values are highlighted in bold).

Domain Shift	RNN-WDCNN	WDCNN	SRDCNN	FFT-SVM	FFT-MLP
**A and B to C**	**99.89 (± 0.13)**	99.68 (± 0.12)	95.24 (± 6.40)	88.99 (± 0.63)	88.60 (± 0.31)
**A and C to B**	**100.00 (± 0.00)**	99.92 (± 0.05)	99.76 (± 0.06)	97.30 (± 0.61)	93.20 (± 0.68)
**B and C to A**	94.98 (± 3.56)	95.28 (± 2.20)	89.22 (± 3.61)	**96.80 (± 0.41)**	94.22 (± 0.37)
**Average**	**98.29 (± 1.23)**	**98.29 (± 0.79)**	94.74 (± 3.36)	94.36 (± 0.55)	92.01 (± 0.46)

**Table 13 sensors-20-05112-t013:** Precision scores of the models considered in this work (as a percentage) for noise rejection performance (best values are highlighted in bold).

SNR	RNN-WDCNN	WDCNN	SRDCNN	FFT-SVM	FFT-MLP
Average	**99.80 (± 0.14)**	96.81 (± 1.15)	78.56 (± 3.96)	92.13 (± 0.30)	90.89 (± 0.33)
None	**100.00 (± 0.00)**	**100.00 (± 0.01)**	**100.00 (± 0.00)**	99.63 (± 0.08)	99.51 (± 0.10)
10 dB	**100.00 (± 0.00)**	**100.00 (± 0.01)**	99.88 (± 0.06)	99.30 (± 0.11)	98.12 (± 0.22)
8 dB	**100.00 (± 0.00)**	99.99 (± 0.01)	99.68 (± 0.16)	98.72 (± 0.18)	97.95 (± 0.11)
6 dB	**100.00 (± 0.00)**	99.99 (± 0.02)	98.44 (± 1.14)	98.29 (± 0.27)	97.13 (± 0.29)
4 dB	**100.00 (± 0.00)**	99.90 (± 0.07)	94.06 (± 2.50)	96.36 (± 0.31)	95.57 (± 0.39)
2 dB	**100.00 (± 0.00)**	98.92 (± 0.52)	81.53 (± 7.57)	94.37 (± 0.32)	93.57 (± 0.37)
0 dB	**99.96 (± 0.06)**	95.56 (± 1.72)	63.90 (± 7.09)	88.83 (± 0.35)	86.96 (± 0.41)
−2 dB	**99.68 (± 0.24)**	91.55 (± 2.37)	48.11 (± 7.06)	81.56 (± 0.47)	79.79 (± 0.43)
−4 dB	**98.60 (± 0.92)**	85.41 (± 5.61)	21.44 (± 10.05)	72.09 (± 0.56)	69.40 (± 0.64)

**Table 14 sensors-20-05112-t014:** Recall scores of the models considered in this work (as a percentage) for noise rejection performance (best values are highlighted in bold).

SNR	RNN-WDCNN	WDCNN	SRDCNN	FFT-SVM	FFT-MLP
Average	**99.79 (± 0.16)**	94.09 (± 1.85)	76.30 (± 3.32)	90.92 (± 0.35)	89.84 (± 0.38)
None	**100.00 (± 0.00)**	**100.00 (± 0.01)**	**100.00 (± 0.00)**	99.63 (± 0.08)	99.51 (± 0.10)
10 dB	**100.00 (± 0.00)**	**100.00 (± 0.01)**	99.88 (± 0.06)	99.27 (± 0.13)	98.09 (± 0.23)
8 dB	**100.00 (± 0.00)**	99.99 (± 0.01)	99.67 (± 0.16)	98.68 (± 0.19)	97.91 (± 0.12)
6 dB	**100.00 (± 0.00)**	99.99 (± 0.02)	98.15 (± 1.59)	98.20 (± 0.31)	97.07 (± 0.32)
4 dB	**100.00 (± 0.00)**	99.90 (± 0.07)	90.86 (± 5.19)	96.09 (± 0.38)	95.35 (± 0.43)
2 dB	**100.00 (± 0.00)**	98.82 (± 0.62)	77.59 (± 5.01)	93.88 (± 0.39)	92.92 (± 0.45)
0 dB	**99.96 (± 0.06)**	93.94 (± 2.77)	60.59 (± 5.91)	87.27 (± 0.53)	84.91 (± 0.74)
−2 dB	**99.67 (± 0.25)**	84.61 (± 5.01)	37.66 (± 6.05)	77.03 (± 0.60)	76.64 (± 0.55)
−4 dB	**98.45 (± 1.13)**	69.59 (± 8.11)	22.31 (± 5.90)	68.20 (± 0.49)	66.17 (± 0.44)

**Table 15 sensors-20-05112-t015:** F1 scores of the models considered in this work (as a percentage) for noise rejection performance (best values are highlighted in bold).

SNR	RNN-WDCNN	WDCNN	SRDCNN	FFT-SVM	FFT-MLP
Average	**99.79 (± 0.16)**	93.63 (± 1.98)	73.29 (± 3.79)	90.71 (± 0.36)	89.62 (± 0.38)
None	**100.00 (± 0.00)**	**100.00 (± 0.01)**	**100.00 (± 0.00)**	99.63 (± 0.08)	99.51 (± 0.10)
10 dB	**100.00 (± 0.00)**	**100.00 (± 0.01)**	99.88 (± 0.06)	99.26 (± 0.13)	98.09 (± 0.23)
8 dB	**100.00 (± 0.00)**	99.99 (± 0.01)	99.67 (± 0.17)	98.68 (± 0.19)	97.91 (± 0.12)
6 dB	**100.00 (± 0.00)**	99.99 (± 0.02)	98.14 (± 1.61)	98.21 (± 0.31)	97.08 (± 0.31)
4 dB	**100.00 (± 0.00)**	99.90 (± 0.07)	89.65 (± 6.64)	96.10 (± 0.38)	95.37 (± 0.43)
2 dB	**100.00 (± 0.00)**	98.81 (± 0.63)	73.43 (± 6.41)	93.93 (± 0.38)	92.95 (± 0.45)
0 dB	**99.96 (± 0.06)**	93.67 (± 2.92)	54.68 (± 6.94)	87.29 (± 0.51)	84.86 (± 0.72)
−2 dB	**99.67 (± 0.25)**	82.66 (± 6.18)	30.03 (± 6.42)	76.46 (± 0.70)	75.91 (± 0.62)
−4 dB	**98.45 (± 1.11)**	67.70 (± 7.94)	14.12 (± 5.84)	66.84 (± 0.59)	64.89 (± 0.47)

**Table 16 sensors-20-05112-t016:** Time costs for training and inference of the models considered in this work (best values are highlighted in bold). Note that the top four models in this table are the variants of RNN-WDCNN considered in the initial experiments in Section 4.2 and Section 4.3.

	Training Times (s)			Inference Times (ms)
	Domain Adaptation Scenario 1	Domain Adaptation Scenario 2	Noise Rejection Scenario	Single Sequence
**LSTM-WDCNN**	3189 (± 8.97)	6389 (± 17.56)	4874 (± 10.54)	3.53 (± 1.56)
**GRU-WDCNN**	3266 (± 7.01)	6554 (± 8.31)	4970 (± 11.62)	3.94 (± 1.04)
**ALSTM-WDCNN**	6792 (± 10.37)	13594 (± 18.66)	10296 (± 13.29)	6.42 (± 1.60)
**AGRU-WDCNN**	5700 (± 13.71)	11389 (± 36.27)	8685 (± 21.86)	5.60 (± 1.37)
**WDCNN**	1144 (± 8.36)	2228 (± 7.05)	1747 (± 6.40)	**1.56 (± 0.76)**
**SRDCNN**	5274 (± 26.39)	10501 (± 11.92)	7978 (± 8.77)	2.21 (± 0.48)
**FFT-SVM**	547 (± 2.69)	1091 (± 1.18)	930 (± 1.64)	10.06 (± 0.34)
**FFT-MLP**	**529 (± 1.53)**	**1030 (± 3.55)**	**890 (± 5.98)**	10.21 (± 0.26)

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
