# Peer review of "A Novel Deep Learning Model for the Detection and Identification of Rolling Element-Bearing Faults"

_sensors, 2020, doi:10.3390/s20185112_

Round 1
Reviewer 1 Report
The authors proposed the RNN-WDCNN deep network-based method to solve the fault diagnosis of rotating machinery. The solution and framework provided by this paper are universal and convincing, but this reviewer did not find the innovation compared to conventional 1D-CNN. This reviewer is concerned about the following points and needs to be discussed with the authors. It is also recommended to further explain these concerns.
-1-. Real-time diagnostic capacity. How to reflect the on-line diagnostic capacity with using this approach? Is that in the testing phase?
-2-. Noise robustness. The authors are recommended to give a further explanation on how to reduce or remove the environmental noise disturbance?
-3-. The database chosen by the authors is very popular. However, this reviewer found in previous verifications that based on this database, some conventional feature extraction methods (time domain-based or time-frequency domain-based methods) combined with some conventional classification tools (such as neural networks, support vector machines) still achieved high recognition accuracy (above 98%). The alternative database, for example, IEEE PHM2012_Festo_Test_set is advised to be added to the test verification of the method proposed by the authors.
-4-. Feature learning and fault diagnosis under variable load operating conditions. In the view of this reviewer, it usually means that the load can be changed continuously, rather than stepwise. This reviewer wanted to know whether the proposed method can solve the problem of fault diagnosis under continuous variable load conditions.
Author Response
Response to reviewers comments – reviewer 1
> We thank the reviewer for the thorough and helpful comments on our work.
Regarding the English language and style, we have thoroughly proof read the manuscript to check the English language usage. Some of the key changes we have made are:
1) The use of third person (particularly in the methodology and results sections)
2) The use of US English (rather than the original UK English)
3) Some spelling and grammar corrections.
These are highlighted in the revised manuscript.
Comments and Suggestions for Authors
The authors proposed the RNN-WDCNN deep network-based method to solve the fault diagnosis of rotating machinery. The solution and framework provided by this paper are universal and convincing, but this reviewer did not find the innovation compared to conventional 1D-CNN.
The novel architecture presented in this work manages to combine some of the best features of conventional 1D CNNs (including their ability to suppress high frequency noise) with the power of recurrent neural networks to capture distant dependencies in time varying signals. The addition of the LSTM pathway allows our proposed approach to capture dynamic temporal features than span large numbers of time steps – thus significantly improving the classification performance over models using a single convolutional pathway. We have highlighted the key contributions of this work (particularly with respect to the strong generalization ability of our proposed approach) in both the abstract and conclusion.
This reviewer is concerned about the following points and needs to be discussed with the authors. It is also recommended to further explain these concerns.
-1-. Real-time diagnostic capacity. How to reflect the on-line diagnostic capacity with using this approach? Is that in the testing phase?
> We briefly discuss the real-time diagnostic capacity in Section 4.4, where we show that the inference time for our RNN-WDCNN model (using an LSTM based recurrent pathway) is around 3.5 ms (on a nVidia GTX 1070 GPU). Each 4096 element vibration sequence represents 85 ms of data (acquired at 48kHz) so, assuming the data acquisition and transmission overheads are low enough, the system can operate online in real-time. We have added discussion of this to the end of Section 4.4 (lines 405 - 408):
“In the data set considered in this work, each 4096 element vibration sequence represents 85 ms of data (with the data acquired at 48kHz). With an inference time of 3.53 ms, this would allow the proposed RNN-WDCNN intelligent fault diagnosis system to process the vibration data online in real-time.”
-2-. Noise robustness. The authors are recommended to give a further explanation on how to reduce or remove the environmental noise disturbance?
> One of the key benefits of our proposed approach is that no preprocessing of the signal is needed to reduce or remove the environmental noise in the vibration data. Our deep learning approach can effective classify bearing fault types even in the presence of large amounts of noise, whereas conventional approaches require noise to removed from the signal using digital filters to operate effectively (as can be seen from the results presented in Section 4.3).
-3-. The database chosen by the authors is very popular. However, this reviewer found in previous verifications that based on this database, some conventional feature extraction methods (time domain-based or time-frequency domain-based methods) combined with some conventional classification tools (such as neural networks, support vector machines) still achieved high recognition accuracy (above 98%). The alternative database, for example, IEEE PHM2012_Festo_Test_set is advised to be added to the test verification of the method proposed by the authors.
> As the reviewer quite rightly points out, there are many conventional classification approaches that are capable of classifying faults in the CWRU dataset with extremely high accuracy (and in fact our proposed approach achieves 100% accuracy when trained and tested on data from the same set of operating conditions – as can be seen in Figure 13, where our RNN-WDCNN architecture achieves 100% accuracy in the presence of no additional noise when trained on all operating conditions). However, the key contribution of this work is that our proposed model is capable of robust generalisation to unseen operating conditions (such as different motor loads, rotating shaft speeds, and environmental noise levels). Previous work in the literature applying conventional approaches to this data set generally focuses on performance when trained and tested under the same set of operating conditions.
We also thank the reviewer for pointing out the IEEE PHM2012 data set. Although the official link to the dataset has been removed from the original website, we have managed to find an archived copy – however, the dataset primarily deals with estimating the remaining useful life of bearings rather than classifying the type and severity of bearing fault (as the CWRU dataset does). Unfortunately the PHM2012 dataset doesn’t contain information about either the nature of the fault or the point where faults start to occur and so is not suitable for use in training the models described in our work. However, predicting the remaining useful life of components is a very interesting area and one that we wish to look at in the future.
-4-. Feature learning and fault diagnosis under variable load operating conditions. In the view of this reviewer, it usually means that the load can be changed continuously, rather than stepwise. This reviewer wanted to know whether the proposed method can solve the problem of fault diagnosis under continuous variable load conditions.
> Our proposed classification approach is capable of operating under continuously varying load conditions as it is capable of robust generalisation to different operating conditions from that on which it has been trained. We have highlighted this contribution in both the abstract and conclusion:
Abstract (line 7):
“The main contribution of this work is the development of an intelligent fault diagnosis method capable of operating on these real-time data streams to provide early detection of developing problems under variable operating conditions.”
Conclusion (line 412):
“The key contribution of this work is the development of an intelligent fault diagnosis framework capable of operating in real-time on unprocessed sensor signals to allow early detection of impending failures under variable load conditions.”
Reviewer 2 Report
Very good research. The analysis is very deep.
Only one question: what your advatage against other deep learning? Why LSTM?
Author Response
Response to reviewers comments – reviewer 2
> We thank the reviewer for the very positive comments on our work.
Comments and Suggestions for Authors
Very good research. The analysis is very deep.
> Thank you for the very positive comments.
Only one question: what your advantage against other deep learning?
> Our proposed approach generalises significantly better to unseen operating conditions and exhibits much better robustness to environmental noise than the current state-of-the-art (including several other deep learning approaches used in the literature).
Why LSTM?
> We examine several candidate recurrent architectures in our work – including LSTM, GRU, LSTM with attention, and GRU with attention – however, the plain LSTM model exhibits slightly better performance in the scenarios we have tested, whilst being faster to train (at least on the hardware used in this work) than the other recurrent architectures considered. The addition of the LSTM pathway allows our proposed approach to capture dynamic temporal features than span large numbers of time steps – thus improving the classification performance over models using a single convolutional pathway.
Reviewer 3 Report
Please see attached file.

Author Response
Response to reviewers comments – reviewer 3
Comments and Suggestions for Authors
> We thank the reviewer for the very thorough and helpful comments on our manuscript and provide below a point-by-point response to these comments. We also provide a highlighted copy of the revised manuscript showing all changes (and provide line numbers in our response below).
Page 1
Need to define it before abbreviations
WD abbreviate what?
> This is now defined in the abstract – we have changed the text to read “recurrent neural
network with wide first kernel and deep convolutional neural network pathway (RNN-WDCNN)”. (See lines 7 - 9)
Page 2
Convolutional Neural Network (as mentioned in part2) FCNN or CNN
> We have changed this to “elements of convolutional neural networks (CNNs)” (line 66)
Page 3
How CNN is used (previously by others) for fault detection and diagnosis?
> The use of convolutional neural networks in fault detection and diagnosis is discussed in Section 2.3.
Utilizing not utilising
> We thank the reviewer for pointing this out, and we have thoroughly proof read the manuscript to ensure that spellings are US English (rather than UK English as previously).
Page 4
Some symbols weren’t define. Please define each symbol
> We have now made sure to define all symbols used in this set of equations (see lines 118 – 119).
Page 5
Is this the same C used in equations 1-5?
> No, this C is the context vector used by the attention mechanism so that the model can learn which elements of the input are most relevant in making an accurate prediction. The c used in equations 1-5 is the activation of the LSTM memory state at time t. We have change the C in this text to bold face (see line 133) and made sure that the c in equations 1-5 is fully defined to avoid confusion.
What do you mean? Shallow and depth in the same sentence do not make sense!?
> We just mean that the convolutional path in LSTM-FCN uses less convolutional stages than many alternative approaches in the literature. We have made this clearer by changing the text to read “the relatively shallow nature of the convolutional pathway (using three convolutional stages)” (see line 164)
Page 6
The proposed. Not our .... rewrite the sentence.
> We have changed this to “The proposed architecture of the recurrent neural network with wide first kernel and deep convolutional path (RNN-WDCNN) model” (see Figure 3).
Symbols in the figure are not defined.
> We have defined some of these in the text and have altered the figure to clarify the meaning of the others (see Figure 3).
Page 7
Define symbols.
> We have defined the meaning of xi and xj in the text (see line 195).
Why only 128? What happens if this number changes?
> Increasing the number of RNN cells here significantly increases the computational complexity of both training and inference using the proposed model; whereas decreasing the number of RNN cells reduces the ability of the model to capture long term dependencies in the input signal. We performed a simple initial hyper-parameter search to determine a good value for this number (as now mentioned in the text) and found that 128 RNN cells provided a good trade-off between computational complexity and classification accuracy. We have clarified this in the manuscript (see lines 198 - 199).
How is the truth vector selected in this work?
> The ground truth vector in this work represents the class of bearing fault, and we have clarified this in the manuscript by adding:
“In this work the ground truth vector represents the one-hot encoded class of the rolling element bearing fault indicated by the input vibration signals.”
(see lines 207 - 209)
Page 9
Not defined in the abstract or anywhere except here. Should be introduced earlier.
> We have now defined this in the abstract (see lines 7 – 9).
Page 11
What happens if the window size decreases or increases ? Is the window size dependent on the shaft speed or the load in this work?
> If the window size decreases it will cover less complete rotations of the motor shaft, and therefore the input signal will contain less relevant information to use in the classification process. In this work the window size is dependent on the shaft speed, and we have chosen a window of 4096 data points to ensure we capture at least 2 complete revolutions of the shaft (and therefore 2 impacts on the bearing fault area). The choice of 4096 data points also makes taking FFTs of the signal more computationally efficient when using conventional machine learning methods in our experiments (though this is a minor consideration).
We have changed the footnote on page 11 to explain this decision:
“A window of 4096 data points was chosen to capture at least 2 complete revolutions of the load bearing shaft - i.e. the rotating motor shaft - and therefore impacts on the bearing fault area. The number of data points in a complete revolution of the load bearing shaft is given by N = Fs × 60 / ω (where N is the number of data points, Fs is the sampling frequency, and ω is the rotating shaft speed in rpm). As Table 4 shows, the rotating shaft speed varies between 1772 rpm and 1730 rpm for the operating conditions investigated in this work. At a sampling rate of 48kHz, this gives between 1626 and 1665 data points per complete revolution which means a window length of 4096 data points is guaranteed to capture at least 2 bearing fault impacts. This approach is similar to that taken by other works in the literature that use the 12kHz sampled data [34,35].”
What is the load bearing shaft are referring to? Is the number chosen is related to the shaft speed, bearing fault characteristic frequency or other parameter(s)?
> The load bearing shaft is the rotating motor shaft. We have clarified this in the revised footnote (see above).
Page 14
Name the two FFT based techniques used. FFT based techniques can only help in identifying periodic signatures, why did you use here? Inner race and ball defects are not periodic faults.
> We have now named the FFT based techniques in Section 4.1.
“compared to two conventional machine learning approaches using Fast Fourier Transform (FFT) based features (one using a multilayer perceptron - FFT-MLP - and one using support vector machines – FFT-SVM)”
(see lines 266 – 268 on page 12 & line 307 on page 13)
We respectfully disagree with the reviewer that ball defects and inner race defects are not periodic. Randall and Antoni (2011) show that for all three types of fault considered in this work periodic impulses in the acceleration signals are produced. For outer race faults these are typically at 1/BPFO (ball pass frequency, outer race); for inner race faults these are typically at 1/BPFI (ball pass frequency, inner race); and for ball faults they are typically at 1/BSF (ball spin frequency) (with additional harmonics at half this period as the ball fault hits both inner and outer race).
Randall, R.B. and Antoni, J., 2011. Rolling element bearing diagnostics—A tutorial. Mechanical systems and signal processing, 25(2), pp.485-520.
Page 16
Rewrite sentence. Must be in third person.
Use other word than whilst. Same word used many times in the manuscript.
> We have fixed both of these in the revised manuscript (and have also thoroughly proof-read the manuscript to find other instances that needed changing to the third person) – see lines 327 – 328.
Page 19
Power? Not poser
> Fixed – thank you for picking this up (see line 366).
Page 20
Why GRU and AGRU methods used here? Can these be also used in the previous test?
> This was a mistake in the headings for the table in Figure 11 (now Figure 12). In this experiment we are comparing the performance of different potential recurrent architectures to use in the RNN pathway (as we did in Section 4.2.1 and Figure 7). We have fixed the table heading and made sure this is explained fully in the discussion of the Figure in Section 4.3.
Page 23
Must mention that time is measured in seconds (s) and milliseconds (ms).
> We have added this in the description of the timing experiments in Section 4.4. This now reads:
“Table 16 shows the average training time (in seconds) for each model on each of the three scenarios considered in Sections 4.2.1, 4.2.2, and 4.3, and the average inference time (in milliseconds) for a single 4096 sample sequence.”
(see lines 391 - 393)
LSTM-WDCNN is the same as RNN-WDCNN? This confusing, must stay with one name and not move back and forth between these names.
> LSTM-WDCNN, GRU-WDCNN, ALSTM-WDCNN, and AGRU-WDCNN are the four variants of the RNN-WDCNN architecture we consider in this work (see Figure 7 and Figure 11). In the scenarios considered in this work the LSTM-WDCNN model performed best out of the four and was thus chosen for use in the comparisons with other algorithms. We have updated both the caption and the layout of table 16 to make this clearer. The caption now reads:
“Time costs for training and inference of the models considered in this work (best values are highlighted in bold). Note that the top four models in this table are the variants of RNN-WDCNN considered in the initial experiments in Sections 4.2 and 4.3”
Why there’s no confusion plots for scenario 2?
> We have added confusion plots for both the domain adaptation scenario 2 (see Figure 11) and the most severe case of additive noise in the noise rejection experiments (see Figure 14). Discussion of these is on lines 345 – 353 and 379 – 386, respectively.
Page 24
Not the correct word to use in this sentence !
> Ablation studies are sometimes referred to in the machine learning literature as a way of studying the performance of a complex model by isolating the functionality of various blocks. However, we appreciate that this may be unclear so have changed the text to read:
“Further work is planned to perform more detailed studies into the effects of the individual elements of our model,”
(see lines 427 - 428)
The authors did show new contributions besides combining more the two methods, what are the main significance of this work in terms of training or classification accuracy.
> By combining a recurrent neural network pathway (with an additional CNN block as a filter) and a deep convolutional neural network pathway we have shown we can achieve better than state-of-the-art results under both varying operating conditions and in the presence of significant amounts of noise.
We have changed the first paragraph in the conclusions section to highlight these key contributions.
“The key contribution of this work is the development of an intelligent fault diagnosis framework capable of operating in real-time on unprocessed sensor signals to allow early detection of impending failures under variable load conditions. The proposed novel dual path deep learning model obtains state-of-the-art classification performance, while also addressing many of the problems with existing intelligent data driven fault diagnosis techniques such as being able to operate on raw time series data (with no need for complex feature engineering) and robustness to noise and changes in operating conditions.”
(see lines 410 - 416)
Many of the references used do not follow the required referencing format. See examples below (highlighted)
> We thank the reviewer for pointing out some of the inconsistencies in the referencing style. We have used the provided LaTeX and BibTeX templates to prepare this manuscript. Unusually, the years for journal publications are in rendered in boldface, and the years for conference publications are rendered in normal typeface. However, thanks to this comment, we have spotted some additional inconsistencies in the references and so have fixed these.
Round 2
Reviewer 1 Report
This reviewer has no more comments.
Reviewer 3 Report
Please edit (fix) the references. It is not following the formatting of the journal. Multiple errors detected in the referencing section.